# Response of Mechanical Properties of Polyvinyl Chloride Geomembrane to Ambient Temperature in Axial Tension

Xianlei Zhang [1,2,*], Zhongyang Ma [1], Yunyun Wu [3,*] and Jianqun Liu [1]

1   School of Water Conservancy, North China University of Water Resources and Electric Power, Zhengzhou 450045, China; mazy2021@163.com (Z.M.); ljq202110100@163.com (J.L.)
2   Engineering Technology Research Center for Safety of Hydro Structure in Henan Province, Zhengzhou 450046, China
3   College of Water Conservancy and Hydropower Engineering, Hohai University, Nanjing 210024, China
*   Correspondence: zhangxianlei@ncwu.edu.cn (X.Z.); 190202020002@hhu.edu.cn (Y.W.)

**Abstract:** Aiming at the mechanical response of geomembrane (GEM) in membrane-faced rock-fill dam (MFRD) to different ambient temperatures, the mechanical properties in axial tension of polyvinyl chloride (PVC) GEM were studied by experiment and theoretical analysis. First, fifteen groups of axial tensile tests for longitudinal/transverse specimens were conducted at different temperatures in the temperature environment laboratory, the stress–strain curve and Young's modulus were obtained, and the variation of Young's modulus with temperature was analyzed by Boltzmann function fitting. Second, the glass transition temperature of PVC GEM was obtained by differential scanning calorimetry (DSC), and the difference in mechanical properties between longitudinal and transverse specimens of PVC GEM was analyzed by thermomechanical analyzer (TMA) thermodynamic test. The results showed that the lower the temperature, the greater the Young's modulus, and the smaller the linear interval of stress and strain, while the higher the temperature, the result is opposite. The difference in mechanical properties between the two directions is related to the ambient temperature. The orientation of polymer structure accounts for the difference in mechanical properties by theoretical analysis. The fitting results of Boltzmann function have a certain reference value for numerical simulation. In design of the membrane impervious structure in MFRD, the ambient temperature should be considered fully, and the longitudinal/transverse welding splicing should be avoided as far as possible. The current test specification should test the mechanical performance of GEM at normal operating temperature of reservoir instead of the test and quality evaluation at a single temperature. The temperature should be considered comprehensively in construction to avoid damaging the performance of impervious structure and ensure the service life.

**Keywords:** Young's modulus; mechanical properties; PVC GEM; ambient temperature; MFRD

## 1. Introduction

GEM is used extensively as an impervious structure in the construction of channels, embankments, dams, reservoir pans, earth and rock cofferdams and water transmission/traffic tunnels because of its flexibility, impermeability, simplicity of construction, easy repairment and cheapness [1–3]. Since GEM was introduced in the 1980s, it has rapidly become one of the most important construction materials in civil engineering; the critical technical problems tackled with research, innovative construction equipment/methods and improved production process all contribute to the development of GEM. Although there is some progress in the application of GEM seepage control technology, there are still some constraints, such as the current design specifications [4,5] restricting the application of GEM seepage control in medium and high dams; the safety and service life of the thinner (≤1.0 mm) GEM are met with skepticism to some extent, etc. [6]; the limitations of the current evaluation for GEM performance; and the complexity of engineering construction.

In addition, the scientific research on GEM seepage control and drainage technology in China primarily focuses on the summary of engineering experience.

The GEM is an organic polymeric material, typically divided into polyethylene, poly (vinyl chloride) and rubber. The polyethylene widely applied in China mainly includes polyethylene membrane (PE), high-density polyethylene membrane (HDPE) and low-density polyethylene membrane (LDPE), whereas rubber is mainly thermoplastic polyolefin (TPO), and poly (vinyl chloride) is mainly soft PVC, and GEM at home and abroad is selectively combined with engineering experience and development, for example, Europe and the United States choose the majority of PVC [7]. Initially, PVC GEM was mainly applied in China, but it gradually replaced by PE/HDPE membrane [8]. In recent years, PVC GEM has been adopted as an impervious material in medium-high dams and dams in deep overburden layer. According to the incomplete statistics by International Committee on Dams in 2010, of 216 large earth-rock fill dams adopting GEM as impervious structure in the world, the number of PVC membrane selected is 143, accounting for approximately 60.3% of the membrane impervious dams, while some dams adopt exposed PVC membrane to prevent seepage [9–11]. PVC GEM still has good flexibility at low temperatures by adjusting the plasticizer type and percentage content, and PE/HDPE GEM fails to maintain the flexibility as that at normal temperatures. The increase in material thickness reduces the flexibility of PE/HDPE, but it has little effect on the flexibility of PVC GEM. Therefore, the widespread use of PVC stems from its strong adaptability to large deformation and good durability.

The research results showed that the principal factor responsible for affecting the mechanical properties of polymer material is the temperature [12–15], and the mechanical properties of GEM present as polymer macroscopical movement, while temperature represents the intensity of molecular motion, and the material exhibits different mechanical properties macroscopically at different ambient temperatures. He, Ping-sheng [16] showed that the time for the response of strain of GEM to external force depended on the ambient temperature and decreases with the increasing temperature. Duttapk [17] suggested that the creep deformation of GEM in the short term at ambient and low temperatures was principally temperature-dependent. V.E. Malpass et al. [18] studied the mechanical properties of PVC membrane at low temperatures. Stark, T.D. et al. [19] tested the rupture strength of PVC GEM joint at 0.6 °C in the field, which was significantly higher than the value in laboratory. Hsuan, Y.G. et al. [20] found that the flexibility of GEM decreased and the brittleness increased at low temperatures. Giroud [21] believed that the changes in ambient temperature caused loss of plasticizers within the PVC GEM; then contraction occurred, and consequently, the flexibility was reduced. Budiman [22] found that the deformation and mechanical properties of GEM depended on temperature by experiment on temperature effects and noted that temperature was crucial in its physical and chemical process. Budiman et al. [23] analyzed the effect of high and low temperature cycling on the brittle failure of GEM. Rowe et al. [24] exposed HDPE membranes to different ambient temperatures and found that the durability decreased with the increasing temperature. Xu, Si-fa et al. [25] found that the temperature stress of HDPE membrane increased linearly with the decreasing ambient temperature. Akpinar, M.V. et al. [26] showed that the temperature affected the interfacial shear strength between GEM and geotextile. Giroud [27] analyzed the relationship between GEM fold height, width, spacing and type and temperature through field test. ASTM D6693/D6693M-04 (Reapproved 2015) [28] considered the effect of temperature on the mechanical properties of GEM and specified the test temperature as 21 ± 2 °C in the Test Procedures for Geosynthetics [29], and the ambient temperature for testing GEM was not strictly required.

Domestic and foreign scholars have studied the effect of temperature extensively; however, the mechanical properties of GEM at different temperatures are rarely studied and not well targeted. The temperature of GEM in engineering practice is not only related to the local air temperature, but also the reservoir water temperature. Particularly, the reservoir water temperature stratifies in high MFRD due to large water depth [30], and

the problems caused by temperature also occur in the construction, such as the maximum temperature of underlying surface being able to reach 40–50 °C during the day, while the temperature decreases at night; the diurnal temperature causes a change in morphology, and then the temperature stress occurs in critical areas such as anchorage. Therefore, the mechanical properties at a single temperature fail to meet the actual needs of engineering. In order to investigate the influence of temperature on the mechanical properties of GEM and provide some reference for the engineering design, the authors focus on the variation of mechanical properties of PVC GEM in axial tension with the ambient temperature.

## 2. Methodology

### 2.1. Apparatus

In order to maintain a constant ambient temperature, the test was conducted in a relatively closed and stable temperature laboratory, which consists of ambient temperature control system and mechanical properties test system for GEM in the extreme environment. As shown in Figure 1, the mechanical properties test system in the extreme environment is mainly composed of servo tester, extreme environment control box, loading control and data acquisition system. The extreme environment control box provides stable ambient temperature for the specimen and performs the axial tensile test in the temperature range of −40 °C to 60 °C combined with the servo tester. The ambient temperature control system consists of a temperature control room and constant temperature laboratory. The former regulates the temperature of constant temperature laboratory with the requirements, and the latter regulates the heat released from the operation of mechanical properties test system in the extreme environment and maintains a stable temperature in the extreme environment control box.

### 2.2. Materials

Commercially available PVC GEM was selected for test. In order to avoid the influence of other impurities such as recycled materials, pure GEM without recycled materials was specially tailored, with a coil width of 2.0 m, a length of 10 m, and nominal thickness of 2.0 mm. The rolls were spread and cut into $1.0 \times 1.0$ m specimen to minimize the effect of curling stress and stored at $20 \pm 2$ °C to avoid mechanical property damage from temperature difference. The mechanical property indexes of the material for axial tension at 24 °C and tensile rate of 5 mm/min were as follows: tensile strength at break 13.79 MPa, tensile strength at yield 3.01 MPa, tensile strain at break 230.18%, tensile strain at yield 58.23%, mass per unit area 1860.5 g/m$^2$, smooth on both sides; all the indexes were measured according to the test procedures for geosynthetics (SL235-2012). The Poisson's ratio of specimen is 0.51 (transverse), 0.49 (thickness direction). According to the GB/T 50290-2014 [31], the tensile strength of the material for longitudinal tension at 24 °C is obtained from the ratio of tensile strength at break to comprehensive strength reduction factor (the maximum of 5 is taken). The main mechanical property indexes of the PVC GEM for testing (at the temperature of 24 °C) are listed in Table 1.

### 2.3. Specimen

The specimen for GEM axial tensile test conforms to ASTM D6693/D6693M-04 (Reapproved 2015) [28] axial tensile test or ASTM D4885-01 (Reapproved 2018) [32] wide strip tensile test, the specimen in the former is dog bone with the ratio of width to length in narrow section W/L = 1:5.5, and it is in a unidirectional tensile stress state without limitation to lateral deformation during tension. As a result, it is mainly applicable to nonreinforced polyethylene and nonreinforced flexible polypropylene GEM, while the specimen in the latter is rectangular, with width W = 200 mm, gauge length L = 100 mm and W/L = 2.0. The necking occurs near the fixture due to the transverse deformation limitation; however, in the central area it is still in a unidirectional tensile stress state, which is suitable for all types of GEM. Merry and Bray [33] conducted quantitively axial tensile tests with specimens of different ratios (W/L = 0.1–5.5), and the results showed that the ratio had little effect on the

stress–strain relationship. Wu, Hai-min [34] used a specimen with a width-to-length ratio W/L = 0.5, gauge length L = 50 mm in order to make the central area of the specimen in a unidirectional tensile stress state, and the test results are more satisfactory. In this paper, the same ratio was used for axial tensile test, with the specimen of width W = 50 mm and gauge length L = 100 mm.

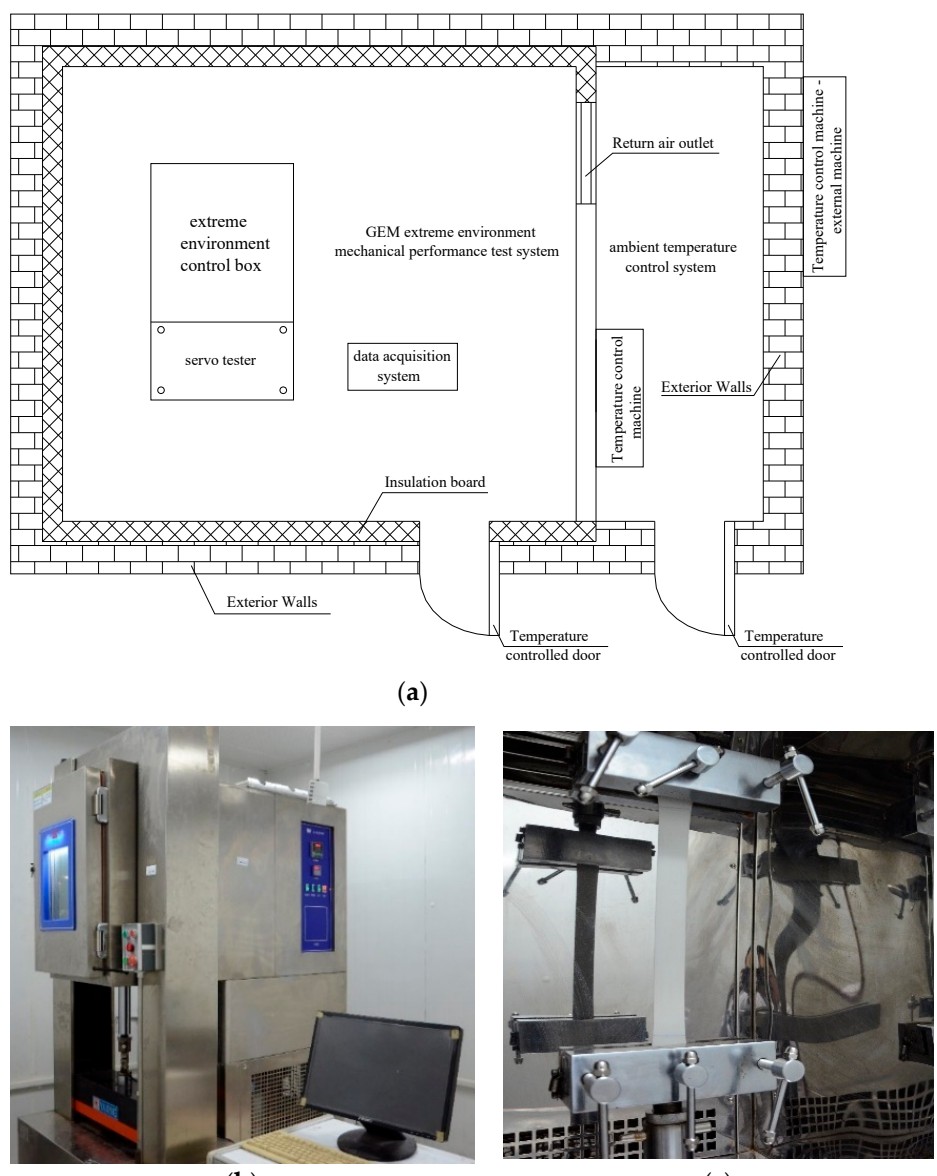

**Figure 1.** Test condition: (**a**) Layout of ambient temperature laboratory; (**b**) Test system for GEM in the extreme environment; (**c**) Loaded specimen.

**Table 1.** Main technical indicators of PVC GEM.

| Technical Indicator | Unit | PVC GEM | Technical Indicator | Unit | PVC GEM |
|---|---|---|---|---|---|
| Average thickness | mm | 2.0 ± 0.5 | Tensile strength | MPa | 2.76 |
| Mass per unit area | g/m² | ≥1800 | Tensile strain at break | % | ≥200.00 |
| Tensile strength at break | MPa | ≥12.00 | Tensile strain at yield | % | ≥50.00 |
| Tensile strength at yield | MPa | ≥2.90 | | | |

### 2.4. Program

The literature [6] classified membrane-impervious earth-rockfill dams into MFRD and core membrane rockfill dam based on the location of GEM impervious structure on the cross section of the dam, and the membrane impervious structure of the MFRD is located on the upstream face of the dam. There exist different temperature zones of membrane impervious structure in reservoir operation. After the reservoir water storage is completed, the temperature of GEM below the dead water level is virtually the same as that at the bottom of reservoir (generally 4–5 °C), which is relatively stable and referred to as constant temperature zone. Affected by profiting regulation, the temperature between the normal water level and the dead water level is variable, the temperature in the range of the reservoir level to the normal water level changes with the air temperature in dam site and that from the dead water level to the reservoir level is related to the temperature of reservoir water. The reservoir level affects classifying the PVC membrane temperature; hence, the GEM temperature in this zone is not only related to the air temperature, but also the reservoir level, and this zone is defined as temperature change B zone. The water level in reservoir operation is basically below the normal water level, and combined with rarely occurred and short checking flood level, the temperature of reservoir water changes slightly; as a result, the temperature of the PVC membrane is in line with local air temperature in the interval from the normal water level to the dam crest elevation, which is defined as temperature change A zone. The working temperature of GEM in MFRD is displayed in Figure 2.

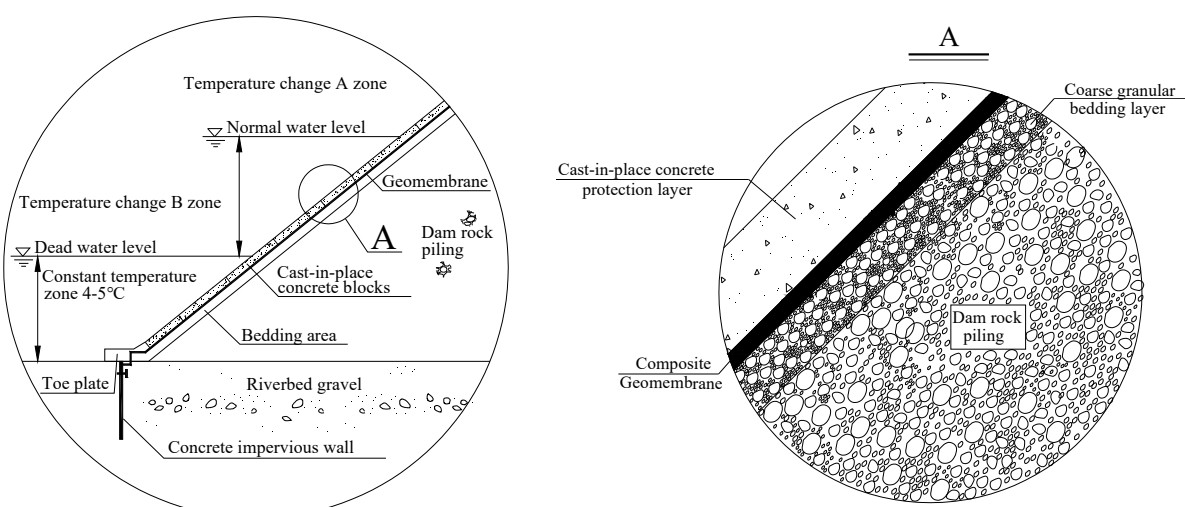

**Figure 2.** Schematic diagram of temperature partition and detail structure of GEM impervious structure of MFRD.

Test temperature design: The ambient temperature of membrane-impervious structure below the dead water level in operation is generally 4–5 °C, because the upper protective layer does not have insulation function. Ignoring its temperature damage, the working temperature of PVC membrane is 4–5 °C; the temperature in A and B zones is associated with the climate temperature in the dam site, and also ignoring the temperature damage of the upper protective layer, that working temperature in A or B zone is consistent with the climate temperature. Considering global built reservoirs, e.g., the record low temperature in the Heihe River Basin in the northernmost part of China reaches −41 °C, while the highest surface temperature in Hainan Island in the southernmost part reaches 55 °C. Therefore, a total of 15 testing temperatures (−40, −30, −25, −20, −15, −10, −5, 0, 4, 10, 20, 30, 40, 50, 60 °C) were selected; the specimens were divided into longitudinal and transverse specimen; and five specimens were prepared at each group with the same temperature.

### 2.5. Procedure

The test procedure is as follows:

(1)　Close the laboratory temperature control door to cut off the heat exchange between laboratory and the external environment, start the laboratory temperature control program, and start the next operation after the laboratory temperature reaches $20 \pm 0.5\,°C$.

(2)　Start the extreme temperature control box, set the test ambient temperature and maintain the space closed; once the temperature reaches the test temperature, install the specimen.

(3)　When the temperature in the extreme temperature control box maintains stable (test temperature $\pm 0.5\,°C$), start the loading control and data acquisition system.

(4)　Stop the test when the tensile displacement reaches the allowable value of the extreme temperature control box (100 mm).

## 3. Results and Preliminary Analysis

### 3.1. Results

The strength performance of geomembrane, e.g., initial tangent modulus, secant modulus and offset modulus are expressed by tensile strength per unit width and corresponding tensile rate in both ASTM D4885-01 (Reapproved 2018) and the test procedures for geosynthetics (SL235-2012). The width used in the specification is the initial width of the specimen; however, the width gradually decreases in the axial tensile process. Therefore, the test methods do have certain limitations. The relationship between axial tension and displacement of the longitudinal and transverse specimens at all the testing temperatures are detailed in Figures 3 and 4, respectively. The studies show that the tensile strain at break of GEM is large (greater than 200%); affected by the internal space of the extreme temperature environment control box, the specimen fails to fracture, whereas the linear range of the relationship between tension and displacement can be obtained, which has no effect on the calculation of the initial modulus. It can be seen from the figure that at the same tensile displacement, the axial tension decreases with the increasing temperature, and the linear range of the relationship between force and displacement increases with the increasing temperature, that is, the higher the temperature, the greater the yield strength, and the smaller the tensile displacement.

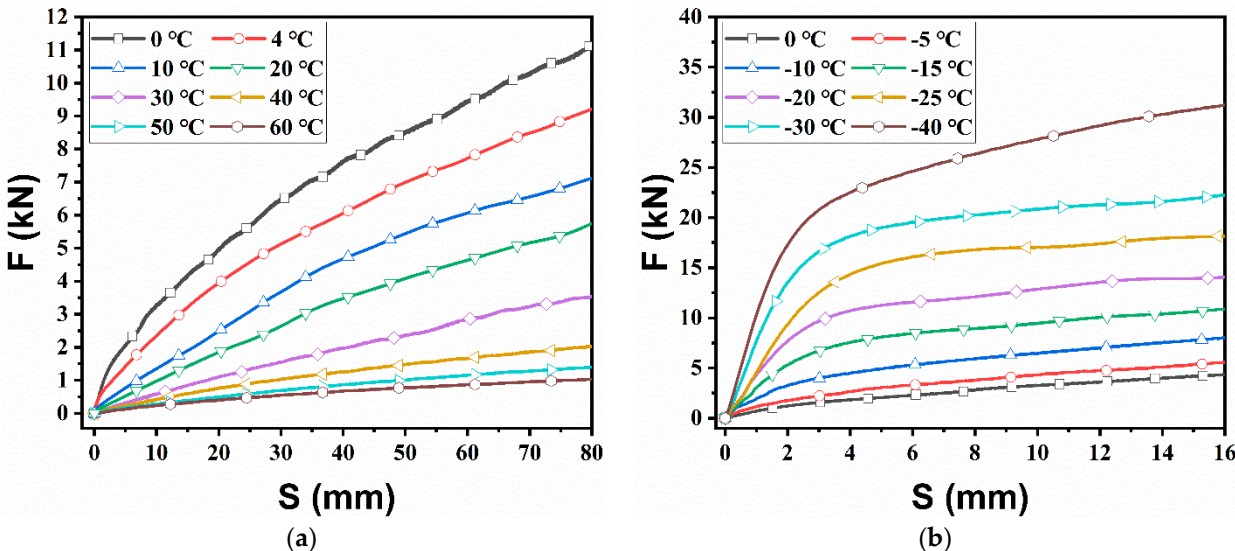

**Figure 3.** Relationship between axial tension and displacement of longitudinal specimen at (**a**) 0~60 $°C$; (**b**) −40~0 $°C$.

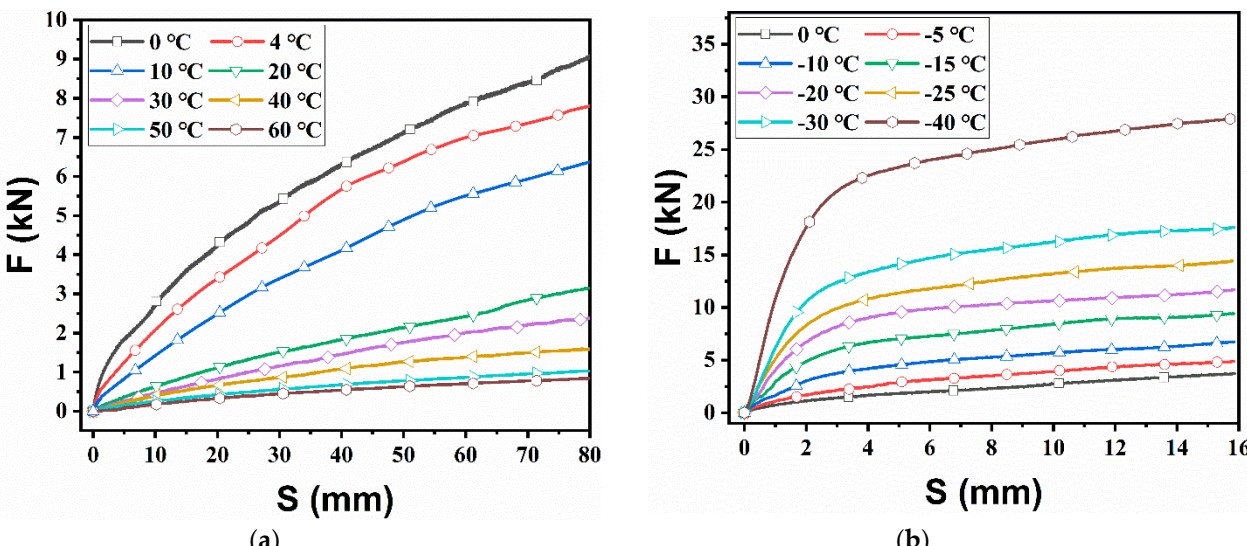

**Figure 4.** Relationship between axial tension and displacement of transverse specimen at (**a**) 0~60 °C; (**b**) −40~0 °C.

Although the variation of mechanical properties of longitudinal specimen with temperature is consistent with that of the transverse specimen, there are certain differences at the same temperature. At the same tensile displacement, the axial tension of longitudinal specimen is higher than that of the transverse specimen. For example, with the tensile displacement of 80 mm at 4 °C, the axial tension of the longitudinal specimen is 11.0 kN, and that of the transverse specimen is 9.0 kN; at −40 °C, the yield strength of the longitudinal specimen is 20 kN, and that of the transverse specimen is 17.5 kN.

### 3.2. Preliminary Analysis

In engineering design for impervious structure of MFRD, the dam height and project level, etc., have different requirements for the impermeability of PVC membranes. The thickness of PVC membranes is selected according to levels of MFRD and affected by other factors. Therefore, the relationship between tension and displacement cannot fully represent the mechanical properties of PVC GEM of different thicknesses.

### 3.2.1. Theoretical Analysis

During the axial tension of the specimen, the axial deformation increases, the transverse and longitudinal (thickness direction) deformation decreases, and the cross-sectional area changes continuously. The relationship between cross-sectional area and axial strain is required to determine the section stress, and the axial strain is defined as:

$$\varepsilon_a = \int_{L_0}^{L_f} \frac{dL}{L} = \ln\left(\frac{L_f}{L_0}\right) = \ln\left(\frac{L_0 + \Delta L}{L_0}\right) = \ln(1 + \varepsilon_{aE}) \tag{1}$$

where

$$\varepsilon_{aE} = \frac{\Delta L}{L_0} = \frac{L_f - L_0}{L_0} \tag{2}$$

The cross-sectional area is given as:

$$A_u = W_{\varepsilon_a} T_{\varepsilon_a} \tag{3}$$

where

$$W_{\varepsilon_a} = W_{\varepsilon_a=0}\left(1 - \mu_y \varepsilon_a\right) \tag{4}$$

$$T_{\varepsilon_a} = T_{\varepsilon_a=0}\left(1 - \mu_z \varepsilon_a\right) \tag{5}$$

Hence, the axial stress may be calculated by the following formula:

$$\sigma = \frac{F}{A_u} = \frac{F}{W_{\varepsilon a=0} T_{\varepsilon a=0} \left(1 - \mu_y \varepsilon_a\right)\left(1 - \mu_z \varepsilon_a\right)} \tag{6}$$

### 3.2.2. Relationship between Axial Stress and Strain

The axial strain and stress can be calculated respectively by Equations (1) and (6) at the testing temperature; the transverse and longitudinal Poisson's ratio can be obtained by three-dimensional digital image processing technology. Based on the test data, Figures 5 and 6 present the stress–strain curves of the longitudinal and transverse specimen, respectively. It can be observed that the axial strain range of the straight section of the stress–strain curve is small at low temperature; for example, the linear range of the longitudinal specimen is [0, 2%] at −40 °C, and that of the transverse specimen is [0, 1%]. At 20 °C, the axial strain range of the straight section of the relationship curve is enlarged, as the linear range of the longitudinal specimen reaches [0, 30%] and that of the transverse specimen is [0, 20%]. With the same axial strain, the lower the temperature, the greater the axial stress.

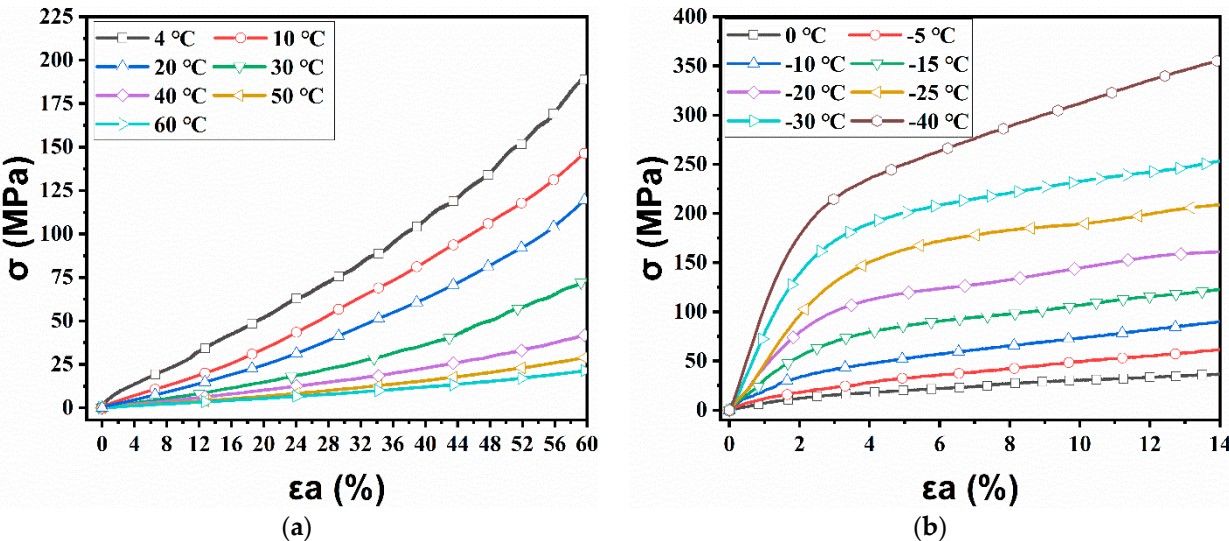

**Figure 5.** Relationship between axial stress and strain of longitudinal specimen at (**a**) 4~60 °C; (**b**) −40~0 °C.

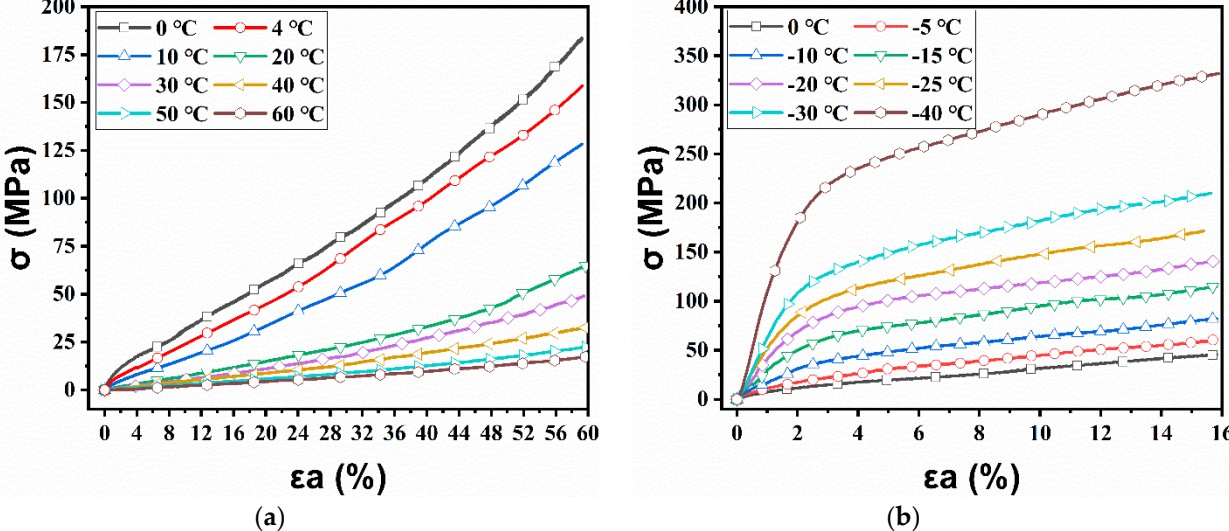

**Figure 6.** Relationship between axial stress and strain of transverse specimen at (**a**) 0~60 °C; (**b**) −40~0 °C.

The stress–strain relationship of PVC GEM varies greatly with the increasing testing temperature, exhibiting a hard and strong form at low temperature and a tough and strong form above room temperature. Therefore, the stress–strain curve of PVC GEM is influenced by the temperature, and the mechanical properties obtained at the temperature specified in the specification do not truly reflect the merits of the material. Combined with the actual temperature in construction and operation, several suitable representative temperatures should be selected for the mechanical properties of the GEM and evaluate the merits of the material to ensure the safe operation of the project.

### 3.2.3. Comparison and Analysis of Mechanical Properties between Longitudinal and Transverse Specimen

Limited by GEM production process, the maximum single membrane width ranges generally from 6 m to 8 m, and adjacent membranes rely chiefly on the welding splicing technology to form an impervious body of the entire dam surface. There is a longitudinal/transverse connection in welding adjacent GEM, and difference in mechanical properties between longitudinal and transverse may affect welding quality due to improper temperature control, resulting in the shorter service life of the structure. Therefore, it is necessary to study the difference in mechanical properties.

Four testing temperatures (−40, −30, 4 and 20 °C) were selected to analyze the mechanical properties of longitudinal/transverse specimen in axial tension, and the comparison results are shown in Figure 7. As can be seen, the stress of longitudinal specimen is higher than that of the transverse specimen. The higher the temperature, the greater the difference in stress between longitudinal and transverse at 4 °C and 20 °C (higher than glass transition temperature), the difference is remarkable at the initial strain and increases gradually with the strain; the difference in stress follows similar patterns at −40 °C and −30 °C, but it is not significant at the initial strain and mainly occurs after the yield point.

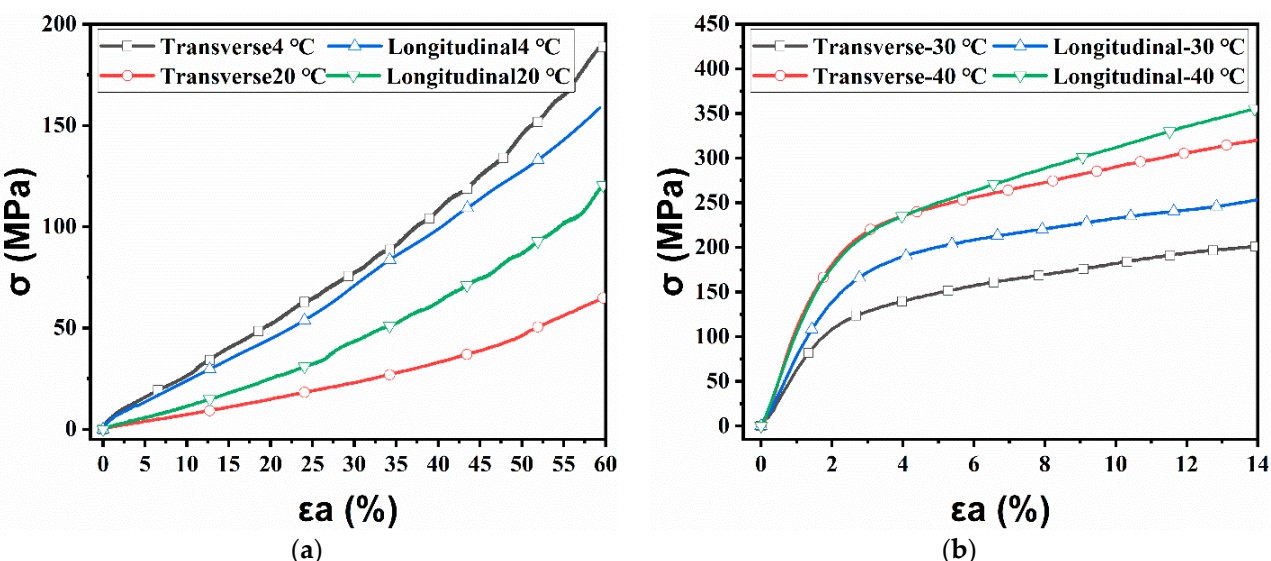

**Figure 7.** Comparison of mechanical properties between longitudinal and transverse specimen at (**a**) 4 and 20 °C; (**b**) −40 and −30 °C.

Figure 7 shows that the longitudinal specimen differs from the transverse in mechanical properties significantly under small strain conditions, and a striking difference is noted with the increasing strain. The GEM is susceptible to tensile failure at the peripheral joints and the uneven settlement on surface, at which the design of GEM impervious structure should consider the directionality of GEM to improve the safety of structure.

## 4. Discussion

### 4.1. Relationship between Young's Modulus and Temperature

The preliminary analysis demonstrates that the stress bears a good linear relationship to strain in the initial tensile stage at each temperature, and the slope of tangent line at linear phase is defined as tensile modulus, i.e., Young's modulus, and the linear intervals of strain are displayed in Table 2. It is evident that the Young's modulus below −10 °C is relatively large, and the variation increases with the decreasing temperature. The Young's modulus is less than 500 MPa above −10 °C, and the variation is small above 4 °C. The linear interval of strain increases gradually with increasing temperature, e.g., the linear interval of longitudinal specimen expands from [0, 2.0%] at −40 °C to [0, 40%] at 60 °C, and that of the transverse specimen expands from [0, 1.0%] at −40 °C to [0, 40%] at 60 °C. The wider linear interval indicates that the PVC maintains good flexibility and adaptability to uneven or local differential settlement of the dam or dam foundation [35].

**Table 2.** Young's modulus and linear interval at different ambient temperatures.

| Temperature | Young's Modulus | Linear Interval | $R^2$ | Young's Modulus | Linear Interval | $R^2$ |
|---|---|---|---|---|---|---|
| | Longitudinal Specimen | | | Transverse Specimen | | |
| °C | MPa | % | — | MPa | % | — |
| −40.0 | 7960.55 | [0, 2.00] | 0.976 | 8014.67 | [0, 1.00] | 0.961 |
| −30.0 | 5975.66 | [0, 2.50] | 0.973 | 5139.42 | [0, 2.50] | 0.979 |
| −25.0 | 3896.48 | [0, 5.00] | 0.972 | 3877.39 | [0, 2.50] | 0.979 |
| −20.0 | 3277.24 | [0, 5.00] | 0.975 | 3336.25 | [0, 5.00] | 0.987 |
| −15.0 | 1990.38 | [0, 5.00] | 0.963 | 1912.44 | [0, 5.00] | 0.966 |
| −10.0 | 469.66 | [0, 10.0] | 0.997 | 1366.01 | [0, 5.00] | 0.993 |
| −5.0 | 391.12 | [0, 10.0] | 0.997 | 492.91 | [0, 5.00] | 0.972 |
| 0.0 | 267.47 | [0, 10.0] | 0.983 | 245.80 | [0, 20.0] | 0.998 |
| 4.0 | 262.70 | [0, 30.0] | 0.999 | 211.97 | [0, 20.0] | 0.994 |
| 10.0 | 154.58 | [0, 30.0] | 0.997 | 151.42 | [0, 20.0] | 0.999 |
| 20.0 | 129.87 | [0, 30.0] | 0.996 | 72.13 | [0, 20.0] | 0.999 |
| 30.0 | 74.26 | [0, 30.0] | 0.989 | 53.54 | [0, 20.0] | 0.999 |
| 40.0 | 55.11 | [0, 40.0] | 0.997 | 45.10 | [0, 40.0] | 0.997 |
| 50.0 | 38.60 | [0, 40.0] | 0.992 | 28.60 | [0, 40.0] | 1.000 |
| 60.0 | 28.54 | [0, 40.0] | 0.998 | 24.42 | [0, 40.0] | 0.998 |

The temperature of GEM impervious structure in construction and operation varies greatly due to climate and reservoir water temperature, so the test for mechanical properties at all temperatures is challenging. If the relationship between Young's modulus and temperature is achieved, then the Young's modulus at the required temperature can be obtained by the test at several concerned temperatures. The Young's modulus versus temperature can be employed in the numerical simulation for MFRD.

In order to further investigate the variation of Young's modulus with the temperature, it is fitted by Boltzmann function, which can be expressed as follows:

$$E = E_2 + \frac{E_1 - E_2}{1 + e^{(T - T_0)/T_r}} \tag{7}$$

The fitting results are described in Table 3 and Figure 8. According to Equation (7), the Young's modulus $E$ infinitely approaches $E_2$ with high enough temperature, whereas $E$ infinitely approaches $E_1$ with low enough temperature. Assuming that the Young's modulus in the molten state is independent of temperature, then the Young's modulus is $E_2$. The measured Young's modulus in initial molten state is 2.01 MPa, while the calculated values of the longitudinal and transverse specimen in Equation (7) are 3.71 and 2.49 MPa, respectively; little difference exists between them. Therefore, $E_2$ can be considered as the Young's modulus in the molten state. As is known, the minimum temperature is

$-273.15\ ^\circ\text{C}$, at which the calculated values of the Young's modulus of the longitudinal and transverse specimen are 9109.85 and 8971.15 MPa, respectively, which are identical with the fitted values. $E_1$ can be considered as the Young's modulus at $-273.15\ ^\circ\text{C}$.

**Table 3.** Fitted values of parameters.

| Specimen | $E_1$ MPa | $E_2$ MPa | $T_0$ °C | $T_r$ °C | $R^2$ — |
|---|---|---|---|---|---|
| Longitudinal | 9109.85 | 3.71 | −30.65 | 10.14 | 0.993 |
| Transverse | 8971.15 | 2.49 | −25.5 | 7.11 | 0.989 |

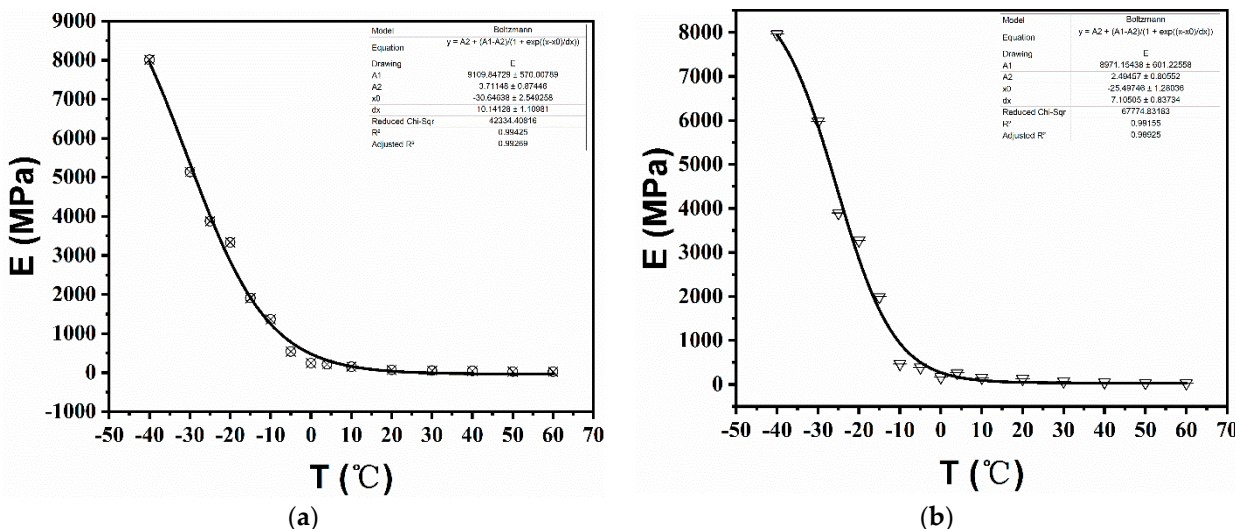

**Figure 8.** Relationship between Young's modulus and temperature: (**a**) longitudinal specimen; (**b**) transverse specimen.

Figure 8 shows that there is a change in the Young's modulus with temperature: a faster rate in the early stage followed by a slower rate. An inflection point of Young's modulus appears with increasing temperature for both longitudinal and transverse specimen, i.e., Young's modulus changes greatly in the interval of $[-40, -9 \pm 2\ ^\circ\text{C}]$, while flat in the interval of $[-9 \pm 2, 60\ ^\circ\text{C}]$. The temperature-dependent polymer has typical glassy state, high elastic state and viscous state, and mechanical properties differ in each state. It can be inferred that PVC GEM is in glassy state and high elastic state in the test temperature range. To confirm the rationality of the inference, DSC was implemented to obtain the glass transition temperature. First, the specimen was melted in the molten state to reduce possible influencing factors; then, the temperature was reduced to $-10\ ^\circ\text{C}$, and finally heated to the molten state, and the DSC curve was obtained. The DSC curve of the specimen at the temperature varying from $-10\ ^\circ\text{C}$ to $10\ ^\circ\text{C}$ is given in Figure 9, from which the glass transition temperature of the PVC GEM is $-7 \pm 0.1\ ^\circ\text{C}$.

The glass transition temperature obtained from the test is close to the opposite number of transverse specimens *Tr* fitted by Boltzmann function. It can be inferred that the opposite number of *Tr* is the glass transition temperature of the PVC GEM, while in contrast, the fitted value of *Tr* for the longitudinal specimen is $10.14\ ^\circ\text{C}$, and its opposite number differs from the glass transition temperature by the DSC test. The reason is that the DSC test is obtained by heating the specimen to the fluid state to eliminate the longitudinal and transverse difference.

Equation (7) can be modified to the following equation for a better physical meaning:

$$E = E_2 + \frac{E_1 - E_2}{1 + e^{(T_0 - T)/T_r}} \tag{8}$$

where the $T_r$ is the glass transition temperature, different from Equation (7).

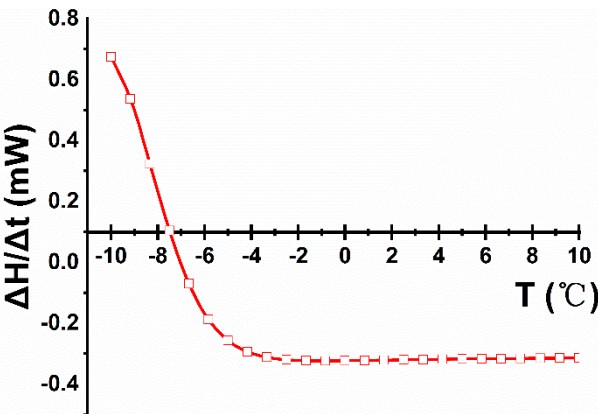

**Figure 9.** Thermodynamic analysis curve by DSC.

### 4.2. Difference in Mechanical Properties between Longitudinal and Transverse Specimen

TMA measures the deformation (one-dimensional dimension) of specimens with temperature under the constant action (non-alternating load), which can heat specimen at a constant rate, so that the specimen deforms with increasing temperature under a constant smaller load, and the temperature-deformation relationship and the contraction and expansion coefficients of the membranes were obtained. In this paper, 5 mm × 10 mm (width × length) longitudinal and transverse specimens were selected for thermodynamic tests, and the results are shown in Figure 10. It is evident that the deformation temperature curves of longitudinal specimen follow different patterns to that of transverse specimen: the deformation of longitudinal specimen is positive and slightly increased at temperatures less than 60 °C; however, it is negative and decreases constantly at a temperature ranging from 60 °C to 150 °C, while that of the transverse specimen is positive and increases with the temperature. The results show that when the ambient temperature exceeds a certain value, the deformation of longitudinal specimen is negative, i.e., heat-absorbing contraction macroscopically, while that of the transverse specimen increases gradually with the increasing temperature and exhibits heat-absorbing expansion.

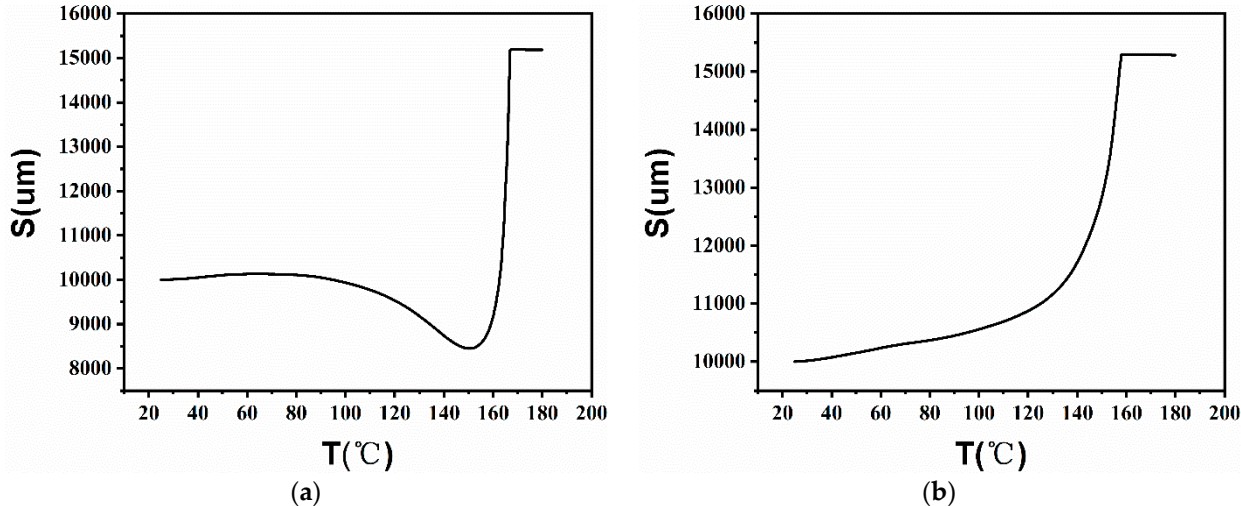

**Figure 10.** Relationship between deformation and temperature: (**a**) longitudinal specimen; (**b**) transverse specimen.

The heat absorption and contraction of the longitudinal specimen is related to the material production process, where the raw material is in the flow state after being fused and mixed and is extruded and condensed at the outlet. The rapid condensed GEM is

pulled to the roller and coiled; molecular orientation appears in the molecular chain by the traction force in condensation molding process. The accumulated deformation by molecular orientation is "frozen" in the internal molecule, and there is no traction effect in the transverse direction [36]. When the temperature activates the polymer motion, the "frozen" deformation begins to return, which is manifested as contraction. The "frozen" deformation is in a tensile stress state, and when the energy produced by temperature is higher than the molecular activation energy, the axial tensile stress needs to offset the stress to be activated before the same displacement increment occurs, resulting in increased tensile stress. As in Table 1, the Young's modulus of the longitudinal specimen at above 0 °C is slightly higher than that of the transverse specimen.

The difference in mechanical properties between longitudinal and transverse specimen is mainly attributed to the production process of GEM. The accumulated deformation of the longitudinal molecular orientation is "frozen" in the molecular structure during the condensation process at the extrusion outlet. The polymer chain is in a tensile stress state. When the energy generated by the temperature action can activate the "frozen" state, the released stress increases the axial force under the condition of displacement-controlled tensile rate, which explains the higher the temperature and the greater difference in Young's modulus between longitudinal and transverse specimen. The difference in the internal stress state accounts for the difference in the mechanical properties of longitudinal and transverse specimen.

## 5. Conclusions

The impervious structure is located on the upstream face of MFRD, of which the GEM is the main impervious material. Ignoring the heat conduction effect of concrete slope above the membrane, 15 tests were conducted on the response of mechanical properties of PVC GEM to ambient temperature in axial tension, and the stress–strain relationship of the longitudinal and transverse specimen was obtained. The difference in the relationship curves was analyzed by theoretical analysis and DSC and TMA thermodynamic test. The results illustrate the variation of the mechanical properties of PVC GEM in response to temperature. Based on the experimental study in combination with the characteristics of GEM impervious structure and the operating environment, the conclusions are as follows:

1.  The stress–strain curve of PVC GEM is greatly affected by temperature, and there is a certain difference between longitudinal and transverse mechanical properties.
2.  The Young's modulus of PVC GEM is related to temperature, and the mechanical properties at a single temperature cannot assess the quality comprehensively, which at the ambient temperature range of the GEM impervious structure in operation can evaluate the quality comprehensively.
3.  Boltzmann function can accurately express the variation of Young's modulus of PVC GEM with the temperature, which has certain reference value for the design and numerical simulation of GEM impervious structure in MFRD.
4.  In the design of critical areas in GEM impervious structure, the potential adverse effects of GEM caused by longitudinal/transverse mechanical property difference should be sufficiently considered to ensure the safety of the structure.

**Author Contributions:** Conceptualization, X.Z.; Methodology, J.L. and Z.M.; validation, Z.M.; formal analysis, X.Z.; data curation, Z.M.; writing—original draft preparation X.Z.; writing—review and editing, Y.W. and X.Z. All authors have read and agreed to the published version of the manuscript.

**Funding:** This research was funded by the National Natural Science Foundation of China (grant numbers 51079047, 51379069). The authors would like to acknowledge the anonymous referees whose comments helped us improve the presentation of this paper.

**Institutional Review Board Statement:** Not applicable.

**Informed Consent Statement:** Not applicable.

**Data Availability Statement:** Not applicable.

**Conflicts of Interest:** The authors declare no conflict of interest.

## Nomenclature

Basic SI units are given in parentheses

| | |
|---|---|
| W | Width of specimen (mm) |
| L | Gauge length of specimen (mm) |
| $\varepsilon_{aE}$ | Axial engineering strain (dimensionless) |
| $\varepsilon_a$ | Theoretical axial strain (dimensionless) |
| $L_0$ | Initial gauge length of specimen (mm) |
| $L_f$ | Gauge length of specimen at the end of the test (mm) |
| $\Delta L$ | Increment of gauge length during axial tension (mm) |
| $A_u$ | Cross-sectional area of specimen during axial tension (mm$^2$) |
| $W_{\varepsilon_a}$ | Width of specimen during axial tension (mm) |
| $T_{\varepsilon_a}$ | Thickness of specimen during axial tension (mm) |
| $W_{\varepsilon_a=0}$ | Initial width of specimen during axial tension (mm) |
| $T_{\varepsilon_a=0}$ | Initial thickness of specimen during axial tension (mm) |
| $\mu_y$ | Transverse Poisson's ratio (dimensionless) |
| $\mu_z$ | Longitudinal Poisson's ratio (dimensionless) |
| $\sigma$ | Axial stress (MPa) |
| F | Measured force during uniaxial tensile test (N) |
| S | Tensile displacement (mm) |
| T | Ambient temperature (°C) |
| $T_r$ | Glass transition temperature (°C) |
| $T_0$ | Fitting constant (°C) |
| E | Young's modulus (MPa) |
| $E_1$ | Young's modulus in the molten state (MPa) |
| $E_2$ | Young's modulus at $-273.15$ °C (MPa) |
| $\Delta H / \Delta t$ | Heat flow rate (mW) |
| $R^2$ | The variance (dimensionless) |

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
