# Peer review of "Response of Mechanical Properties of Polyvinyl Chloride Geomembrane to Ambient Temperature in Axial Tension"

_applsci, doi:10.3390/app112210864_

Round 1

Reviewer 1 Report

Reference: Appl. Sci. 1454839

Title: Response of Mechanical Properties of PVC Geomembrane to Ambient Temperature in Axial Tension

The research concerns, of the mechanical properties in axial tension of PVC GEM  studied by experiment and theoretical analysis. Fifteen groups of axial tensile tests for longitudinal/transverse specimens were conducted at different temperatures in the temperature environment laboratory, the stress-strain curve and Young's modulus were obtained, and the variation of Young's modulus with temperature was analyzed by Boltzmann function fitting. Additionally, the glass transition temperature of PVC GEM was obtained by DSC and the difference in mechanical properties between longitudinal and transverse specimen of PVC GEM was analyzed by TMA thermodynamic test. The results showed that the lower the temperature, the greater the Young's modulus, and the smaller the linear interval of stress and strain, while the higher the temperature, the result is opposite. The difference in mechanical properties between the two directions is related to the ambient temperature. The orientation of polymer structure accounts for the difference in me-chanical properties by theoretical analysis. The results were summarized on the basis  of  experiments, possessing a certain degree of innovation. However article needed to be improved.

  1. List the mechanical property indexes of the material for axial tension in the table (2 Materials).
  2. Why the mechanical property indexes of the material for transverse tension for 24°C were not given? Describe in the table (2 Materials).
  3. Can you provide data from the literature what is the glass transition temperature of PVC? What could affect the glass transition temperature of PVC GEM (lowering of the glass transition temperature of PVC GEM).

Author Response

Dear reviewer:

We have revised the manuscript according to your opinion, please see the attachment. 

Thank you.

Reviewer 2 Report

Good paper, valid for people working with geomembranes, clearly explained (Introduction) why PVC is more often used than PE or HDPE. Broad range of performed experiments - page 3 - in temperatures ranging from -40oC to 60oC. I would like to suggest to the Authors to add few sentences of their opinion - did they observe loss of plasticizer from PVC GEM ? Did they know something about aging resistance of PVC GEM ? Did they measure the Tg value of used PVC GEM ? What is their opinion comparing PVC and PE, HDPE geomembranes ?

There are few editorial mistakes, easy to correct but irritiating to the reader:

page 2 - two times should be ...poly(vinyl chloride)...

page 2  - why MALPASS is writtten in capitals ?

page 3 - there should be added space between numerical value and units, for example it should be written ....2.0 m; 10 m; 2.0 mm; 13.79 MPa; 3.01 MPa; 1860.5 g/m2

Page 10 - there should be added space between numerical value and units, for example it should be written ....2.01 MPa; 8971.15 MPa; 20 kN

Author Response

(The authors gave the same response as above.)

Reviewer 3 Report

  • “GEM is used extensively as an impervious”
  • “such as: (delete “:”) the current design specifications [4-5] restrict the application of GEM
  • [12-15], and the mechanical properties of GEM presents (delete “s”) as polymer movement macroscopically”
  • “as polymer movement microscopically” → "as polymer microscopic movement"
  • “He, Ping-sheng [16] showed that the time for the response of strain of GEM”
  • “GEM in the extreme environment”
  • “in the temperature range of -40℃ to 60℃ combining (correct to combined)”
  • “however, (put the comma) in the central area it is still in a unidirectional tensile stress state’
  • “that of the transverse specimen, there are certain differences at the same temperature”
  • “decreases constantly at a temperature ranging from 60℃ to 150℃, while that of the transverse specimen is positive and increases with the temperature”
  • “15 tests were conducted on the response of mechanical properties of PVC GEM to ambient temperature in axial tension, and the stress-strain relationship of the longitudinal and transverse specimen was (change were to was) obtained”
  • increases with the temperature”
  • “15 tests were conducted on the response of mechanical properties of PVC GEM to ambient temperature in axial tension, and the stress-strain relationship of the longitudinal and transverse specimen was (change "were" to 'was") obtained”
  • 3.2.3 - indicate why such temperature values (-40, -30, 4, 20 °Ð¡) were taken for analysis
  • how the temperature deviation was determined [-40, -9 ± 2 ° C] when you calculated Young's modulus? (page 10 from14)
  • how the temperature deviation was determined [-9 ± 2.60 ° C] when calculated in Young's modulus? (page 11 from14)
  • The nomenclature does not contain explanations of the letters used throughout the text of the article, namely: R, L, W, S, dH, dt
  • Figure 9 contains letters "dH/dt", maybe they should be designated as "ΔH/Δt?
  • There are no decryptions for abbreviations PVС, DSC, TMA at the first mention in the abstract of the article

Author Response

(The authors gave the same response as above.)

Reviewer 4 Report

The paper presents an interesting study of the mechanical response of geomembrane(GEM) in membrane faced rockfill dam
(MFRD) to different ambient temperatures.

However, there are certain areas of the paper that could be improved. The set up testing and laboratory program should be improved with a more clear visualization of the samples different conditions and also of the testing procedure.

At the conclusion part, the engineering recommendation based on temperature variations and usage of geomembrane(GEM) in practice are not very clear.

Author Response

(The authors gave the same response as above.)
